# Augmenting CT-Guided Bone Biopsies Using ^18^F-FDG PET/CT Guidance

**DOI:** 10.3390/cancers16152693

**Published:** 2024-07-29

**Authors:** Max F. Droste, Floris H. P. van Velden, Matthias N. van Oosterom, Valentijn J. Luijk, Mark C. Burgmans, Tessa Buckle, Fijs W. B. van Leeuwen, Daphne D. D. Rietbergen

**Affiliations:** 1Section of Nuclear Medicine, Department of Radiology, Leiden University Medical Center, 2333 ZA Leiden, The Netherlands; m.f.droste@lumc.nl (M.F.D.); f.h.p.van_velden@lumc.nl (F.H.P.v.V.); 2Interventional Molecular Imaging Laboratory, Leiden University Medical Center, 2333 ZA Leiden, The Netherlands; m.n.van_oosterom@lumc.nl (M.N.v.O.); v.j.luijk@lumc.nl (V.J.L.); t.buckle@lumc.nl (T.B.); f.w.b.van_leeuwen@lumc.nl (F.W.B.v.L.); 3Section of Interventional Radiology, Department of Radiology, Leiden University Medical Center, 2333 ZA Leiden, The Netherlands; m.c.burgmans@lumc.nl

**Keywords:** image-guided bone biopsy, ^18^F-FDG PET/CT, CT-guided biopsy, bone lesions, histopathology

## Abstract

**Simple Summary:**

Computer tomography (CT)-guided percutaneous core biopsies are currently the gold standard in diagnostic procedures for patients with bone lesions of unknown kind. Morphologic targeting can be challenging, especially for small and/or heterogeneous lesions. Hereby, targeting inaccuracies may lead to misdiagnosis or inconclusive histopathology. This increases the need for repeat biopsies, which is associated with an accompanying increase in healthcare costs, delay in diagnosis and treatment, and decrease in quality of life. Interventional molecular image guidance has the potential to refine lesion localization and personalize the biopsy strategy. By using 2-deoxy-2-^18^F-fluoro-D-glucose-positron emission tomography (^18^F-FDG-PET)/CT to support the identification of heterogeneous lesions or lesions without radiographic substrate, we show that molecular image guidance helps to increase the procedural success rate by 16%.

**Abstract:**

Computer tomography (CT)-guided percutaneous core biopsies are currently the gold standard in diagnostic procedures for patients with bone lesions of unknown kind. CT-guided biopsies can lead to misdiagnosis or repetition of biopsies in case of small or heterogeneous lesions. We hypothesize that molecular image guidance could be used to optimize the biopsy strategy, by supporting the detection of heterogeneous lesions or lesions without radiographic substrate. To evaluate this hypothesis, we investigated if and how the addition of 2-deoxy-2-^18^F-fluoro-D-glucose-positron emission tomography (^18^F-FDG-PET)/CT could augment routine CT-guided bone biopsies. To this end, 106 patients who underwent a CT-guided bone biopsy between April 2019 and April 2020, obtained from either a vertebral or peripheral bone, were included. Patients were divided into 2 groups: 36 patients received an ^18^F-FDG-PET/CT scan prior to their CT-guided bone biopsy (PET group), while 70 patients only had a morphological CT scan (CT group). Histopathology was used to categorize biopsies into five subgroups (inconclusive, benign, malignant or infectious disease, or normal tissue). In the PET group, the number of conclusive biopsies was significantly higher compared to the CT group (N = 33/36 (92%) versus N = 53/70 (76%); *p* < 0.05). Furthermore, the number of first-try biopsies was lower in the PET group compared to the CT group (1.9 vs. 2.54, *p* = 0.051). In conclusion, ^18^F-FDG-PET/CT imaging significantly increased the success rate of first-try CT-guided bone biopsies by showing less inconclusive biopsies and misdiagnosis.

## 1. Introduction

Bone abnormalities are presented in different ways before diagnosis, e.g., on the basis of patients’ complaints, ancillary findings, or during dissemination and staging. Most of the bone abnormalities are found following radiological imaging in the form of conventional X-ray examination or computed tomography (CT) imaging. To achieve more adequate staging of benign and malignant lesions, morphological data can be supplemented with molecular imaging [1,2,3,4]. In the last decades, positron emission tomography (PET)/CT scans using the glucose analogue 2-deoxy-2-^18^F-fluoro-D-glucose (^18^F-FDG) have been included in numerous cancers and inflammation guidelines [1,2,3,4]. During such a PET/CT scan, the patient is imaged from the skull to mid-femoral or toes, thereby enabling the detection of additional (distant) lesions.

Still, for a definitive diagnosis of bone lesions, histopathology, and not imaging, is the gold standard. Histopathology relies on biopsies, sampling of which tends to be guided by established CT guidance protocols [5]. Adequate histopathology and immunopathology lead to adequate diagnosis, better staging, and adaptation of treatment options [6]. However, CT-guided biopsies do have their limitations. The degree of inconclusive histopathology or even misdiagnosis depends on lesion sizes, morphologic characteristics on radiographic imaging, lesion heterogeneity, and the differentiation grade. In case inconclusive biopsy results occur, additional (i.e., repeat) biopsies are needed. Such repeat procedures are associated with increasing healthcare costs, delays in diagnosis and treatment, and decreases in quality of life [7].

As ^18^F-FDG-PET/CT scans can potentially help overcome the limitations of CT-guided biopsies, e.g., in case of absence of anatomic substrate on CT imaging and by providing information on heterogeneity [8,9], these scans could provide synergistic value to existing CT-guided biopsy paradigms [10]. To test this hypothesis, we performed a retrospective cross-sectional study to evaluate if and how incorporation of ^18^F-FDG-PET/CT guidance increases the accuracy of CT-guided percutaneous bone core biopsies and decreases the number of inconclusive biopsies.

## 2. Materials and Methods

### 2.1. Patient Demographics

In this cross-sectional study, all patients who underwent pathohistological-confirmed CT-guided bone biopsy, either from a vertebral or peripheral bone, at the department of Radiology in the Leiden University Medical Center between April 2019 and April 2020 were retrospectively included and analyzed. Patients were only included when (1) the histopathology report was present, or (2) the difference in time between the ^18^F-FDG-PET/CT scan and the CT-guided biopsy did not exceed 75 days. This time difference was arbitrary and was chosen as a precaution so that the molecular signal still matched the biopsy.

The patients who underwent pathohistological-confirmed CT-guided bone biopsy were divided into two groups based on the availability of a ^18^F-FDG-PET/CT scan prior to biopsy, resulting in a PET group and a CT group (Figure 1). Gender, age, length, weight, indication (primary malignant, metastatic disease, benign, or infection), and location of bone biopsy were documented for all patients. The number of biopsies (first and repeat) was scored, and all biopsies were accompanied by a histopathologic analysis report. Both groups (PET and CT groups) were divided in 5 subgroups, based on histopathologic disease findings (inconclusive findings, benign, malignant or infectious disease, and normal tissue). This retrospective study was evaluated by the Medical Ethical Review Board of the department of Radiology in the Leiden University Medical Center and a waiver was obtained for retrospective use of the patient data.

### 2.2. Imaging Procedures

#### 2.2.1. Conventional Imaging and CT Imaging

Prior to biopsy, all patients received conventional X-ray (Triathlon T3, Odelft, Delft, Benelux) or CT imaging (Aquilion One, Canon Medical Systems, Otawara, Tochigi, Japan), depending on the way they presented themselves at their referring doctor, e.g., on the basis of patients’ complaints, ancillary findings, or during dissemination/staging, according to the guidelines.

#### 2.2.2. PET Group

A subgroup of the patients received molecular imaging with ^18^F-FDG-PET/CT (PET group) according to the guidelines of the suspected disease (e.g., to diagnose or stage malignancies). ^18^F-FDG-PET/CT images were acquired in accordance with the European Association of Nuclear Medicine (EANM) guidelines for tumor PET imaging [11] using a Vereos PET/CT (Philips Healthcare, Best, The Netherlands). All patients fasted for at least 6 h and serum glucose levels were below 8.0 mmol/L. PET acquisition was started 60 min (range 55–75 min) after intravenous administration of ^18^F-FDG using a quadratic dose scheme according to the formula (MBq): 379 × (patient weight (kg)/75)^2^/emission acquisition duration per bed position (min·bed^−1^). Prior to the PET scan, a low-dose CT scan (120 kVp, 50 mAs) was acquired for attenuation correction purposes and anatomical reference. The reconstructed voxel sizes were 4 × 4 × 4 mm^3^ for PET and 1.17 × 1.17 × 5 mm^3^ for the low-dose CT images.

After acquisition, all PET/CT images were evaluated (visually) by an experienced nuclear medicine physician (D.D.D.R.) with over 13 years of expertise in interventional nuclear medicine, with respect to localization, ^18^F-FDG uptake, tumor size, tumor heterogeneity, and the corresponding morphologic substrate on CT. The PET/CT results were interpreted independently of the histological report.

#### 2.2.3. CT-Guided Biopsy Procedure

All patients were subjected to the same local CT-guided biopsy procedure: sterilization of the skin with 0.5% chlorhexidine, use of sterile drapes to create a sterile field, and sedation of the biopsy side with 10 mL of Lidocaine 2% and 2 mL of sodium bicarbonate 8.4%. When anesthesia was needed, 5–10 mg of diazepam i.v. was induced for muscle relaxation and light sedation.

Prior to the guided biopsy, a CT scan (Aquilion One, Canon Medical Systems, Otawara, Tochigi, Japan) was performed to define the optimal biopsy site and needle path. Anatomical landmarks were used for localization of the lesion during guiding. CT-guided biopsies were performed with either Jamshidi 8 G or 11 G or Bonopty 12 G or 14 G sets (Apromed, Ostrava, Czech Republic), including a penetration set and extended drill. The needle procedure was performed after making a small incision in the skin of the biopsy site for needle insertion. When the biopsy needle was inserted, a second CT acquisition was acquired to assess the needle placement. When fluoroscopy indicated a good placement of the needle, the biopsy was taken. At least one biopsy was obtained for each patient. For some patients, multiple biopsies were obtained from the same lesion. After the biopsy procedure, the skin was stitched, and a pressure bandage was applied. Molecular input from the PET scan was available for interpretation or needle placement and was not used on a structural basis for biopsy planning, only employed at the convenience of the radiologist.

#### 2.2.4. Histology

Biopsies were placed on a sterile non-woven gauze and moisturized with NaCl. Afterwards, they were fixed in formaldehyde and sent to a pathologist for histopathological analysis. All biopsy samples were classified by an experienced pathologist dedicated to the specific disease type/indication and categorized by histopathologic findings (e.g., tumor cells, normal bone stroma, and the absence of representative material) in 5 subgroups, i.e., either conclusive (benign, malignant, infection, or normal tissue) or inconclusive.

### 2.3. Statistical Analysis

The statistical analyses were performed using SPSS (version 25.0; IBM, Chicago, IL, USA). All numerical values were first assessed on nominal distribution with a Shapiro–Wilk test. All variables were not normally distributed and were, therefore, analyzed with the non-parametric Mann–Whitney U test. The group comparison of categorical values was analyzed with a Pearson’s chi-square test. *p*-values less than 0.05 were considered significant.

## 3. Results

### 3.1. Included Patient Data

In this study, 112 patients were treated within the timeframe of analysis. For 111 patients (median age: 63 years, range: 37–75 years), corresponding histopathologic confirmation, either from a vertebral or peripheral bone, was provided. For a further five patients, the time between ^18^F-FDG-PET/CT imaging and the CT-guided biopsy procedure exceeded 75 days; in one patient, the pathology report could not be retrieved. Patients who did not fulfill the inclusion criteria were excluded from further analysis (Figure 1), resulting in inclusion of 106 patients (N = 36 in PET group and N = 70 in CT group). The indication for the performed biopsy was categorized into four subgroups: primary malignant, benign, metastatic disease, or infection (Table 1). Patients’ characteristics are displayed in Table 1.

### 3.2. PET versus CT Group Prior to CT-Guided Biopsy

A significant difference was seen in the biopsy indication between both groups (*p* = 0.02). The PET group primarily included malignant tumors (83%, either primary malignant or metastatic disease). In the CT group, 30% of the biopsies had a primary malignant indication, 29% metastatic disease, 24% a benign indication, and 17% were performed to confirm or rule out infectious disease (Table 1).

There were no significant differences in baseline patient characteristics (gender, age, length, or weight) between both study groups (Table 1). The average number of biopsies counted for each patient was 1.9 in the PET group and 2.54 in the CT group (*p* = 0.051), indicating a trend. An overall significant difference was seen in the location of the biopsied lesions (*p* = 0.008; Table 1). More biopsies were performed in the pelvic area in the PET group compared to the CT group (56% and 23%, respectively; *p* = 0.001). Compared to the PET group, the CT group contained more biopsies of lesions located in the lower extremities (tibia) and spine (6% and 21%, respectively; *p* = 0.035). Concerning the histopathologic outcome, a significant difference was seen in primary tumor diagnosis in the CT group (47% and 22%, respectively; *p* = 0.013), and for the metastatic disease, a significant difference was seen in the PET group (67% and 22%, respectively; *p* < 0.001).

### 3.3. Diagnostic Performance and Histopathologic Findings

All biopsies were classified by the pathologist as either conclusive or inconclusive, and the conclusive diagnosis was categorized into four subgroups (benign, malignant, infection, or normal tissue; Table 2). This yielded a significant difference between the PET and CT groups with respect to the conclusiveness of the biopsy (n = 33 vs. n = 53). This difference indicated that ^18^F-FDG-PET/CT yielded a significantly higher (*p* < 0.05) number of conclusive biopsies, from 76% to 92%. We also observed a three-fold decrease (*p* < 0.05) in the inconclusive rate for PET-guided biopsies compared to CT guidance alone (8% vs. 24%). When subdividing these results into histopathologic categories, a significantly increased number of malignant confirmations was seen in the PET group (69% vs. 36%, respectively; *p* < 0.001). Moreover, a significantly increased number of benign confirmations was seen in the CT group (8% vs. 26%, *p* = 0.03). In both the PET and CT groups, there were five biopsies showing normal tissue as a result (14% vs. 7%). No comparison was possible for the indication of infection, as none of these patients underwent a ^18^F-FDG-PET/CT.

### 3.4. Repeat Biopsy

In 17% of the inconclusive cases, the biopsy had to be replanned: five patients underwent a second CT-guided biopsy and seven patients underwent an open biopsy in the operating suite. Only one patient received a ^18^F-FDG-PET/CT prior to the second biopsy, and in this second attempt, pathology confirmed malignant disease. Of the second-try biopsies (CT-guided or open), 27% resulted in a proper diagnosis; in the other patients, the results remained unchanged.

### 3.5. Clinical Examples

Figure 2 illustrates an example of a biopsy for a patient diagnosed with non-small-cell lung cancer (NSCLC), where ^18^F-FDG-PET/CT revealed a suspicion of oligometastatic disease in the os ilium. On CT, a lytic lesion was seen in the left os ilium. Fused PET/CT showed only metabolic activity in the peripheral zone, where the core of the lesion hardly showed any activity. For biopsy, the tip of the needle was placed more in the dorsal part instead of the ventral or central parts.

Figure 3 shows the added value of metabolic imaging using ^18^F-FDG-PET/CT in a patient with B-cell lymphoma. The ^18^F-FDG-PET images revealed increased uptake in the medial condyle of the left distal femur and slightly in the lateral condyle of the distal femur, but the CT images did not show any morphologic substrate of the metabolically active lesion. In this case, ^18^F-FDG-PET/CT was able to accurately guide the percutaneous bone biopsy.

Figure 4 shows the added value of metabolic imaging using ^18^F-FDG-PET/CT in a patient with a primary bone tumor. In the first attempt, the needle was placed in the peripheral zone of the femur. The histopathologic outcome revealed inconclusive findings. A second try was performed after receiving the metabolic information. In the left image, the tip of the needle is clearly positioned in soft tissue, with increased uptake on the performed pre-interventional metabolic imaging. When comparing the needle path between biopsies (Figure 4), it becomes evident that the angle and location of the tip of the needle are different, resulting in a different target area.

## 4. Discussion

CT-guided percutaneous core biopsies are currently the gold standard in diagnostic procedures for bone lesions of unknown origin. Unfortunately, bone biopsy based on CT guidance may cause a delay in diagnosis, proper staging, and treatment of disease [12,13]. Our findings indicated a significant 1.2-fold (16%) increase in successful biopsy rates and a 3-fold (16%) decrease in inconclusive rates when metabolic ^18^F-FDG-PET/CT was included. Furthermore, the inclusion of ^18^F-FDG-PET/CT significantly reduced the number of performed biopsies on the patient base.

To the best of our knowledge, it has not yet been investigated whether or not complementary ^18^F-FDG-PET/CT imaging would be a valuable addition to conventional CT-guided biopsies. That said, multiple studies have been conducted comparing morphologic CT- and metabolic ^18^F-FDG-PET/CT-guided techniques [14,15,16,17]. In some cases, software was used for co-registration of both imaging modalities and/or assisted by virtual reality for biopsy guidance [10,14,18,19]. Studies using ^18^F-FDG-PET/CT for the characterization of the lesion of interest reported that only a portion of a bone lesion is malignant and a suitable location for a biopsy, thus evading benign tissue as fibrosis or inflammation [20,21,22,23]. By exploiting the property of ^18^F-FDG-PET/CT to accumulate in the metabolically most active regions, Klaeser et al. [24] found a way to avoid a necrotic biopsy. This was also confirmed by the study of Purandare et al. [16], stating that the addition of ^18^F-FDG-PET/CT showed an improvement when dealing with heterogeneity in lesions and avoiding necrotic core biopsies. Furthermore, Wang et al. [25] stated that the addition of ^18^F-FDG-PET/CT as an imaging modality creates a reliable means of classifying and diagnosing a bone lesion. Cornelis et al. [26] showed that ^18^F-FDG-PET/CT had an additional value (sensitivity of 100%) when other imaging modalities only showed poor visualization of the lesion. They, however, indicated that false positives may be a problem. The 100% sensitivity and positive predictive value of ^18^F-FDG-PET/CT in biopsy guidance was also mentioned in studies by Cornelis et al., Cerci et al., and Guo et al. [19,26,27], indicating that ^18^F-FDG-PET/CT qualifies as a good biopsy-guiding imaging modality. In their studies, the first-try success of a ^18^F-FDG-PET/CT-guided biopsy was 94.3–96.8%, similar to the 91.7% observed in our present study (at most, 5.7% inconclusive biopsies, similar to the 8.3% of inconclusive biopsies observed for the PET group in our present study). If we compare this to the CT-guided biopsies showing a first-try success between 61% and 98% (75.7% observed in our present study), the addition of ^18^F-FDG-PET/CT could lead to a better diagnostic tool and more patient-friendly approach [9,19,28,29,30,31,32,33]. In addition, tumors can be heterogeneous, in which some parts will have a higher degree of malignant differentiation than other parts. Especially in the era of molecular profiling and therapy options based on tumor mutations, the detection of the highest malignancy rate will become increasingly important.

Although promising results were found in our study, there are also a few limitations. Due to the retrospective nature of this study, there is uncertainty regarding the way the interventional radiologist implemented the information from the ^18^F-FDG-PET/CT to guide the biopsy needle. Real-time PET/CT biopsies are currently not performed due to radiation exposure issues. For ^18^F-FDG-PET/CT-guided interventions, this could potentially be resolved by several means. To achieve the lowest possible personnel dose, a robotic arm can be used to insert a biopsy needle into the patient, while maintaining the highest possible diagnostic success in live ^18^F-FDG-PET/CT-guided percutaneous core biopsies [34,35]. Using isotopes with a longer physical decay, such as ^89^Zr, may help overcome logistical challenges, but will result in a higher patient-absorbed dose [36,37]. New-generation, state-of-the-art PET/CT systems are becoming more and more available on the market, with a higher sensitivity and better reconstruction techniques, e.g., those based on artificial intelligence, using a reduced radiopharmaceutical dose to decrease radiation exposure for patients and medical staff. Secondly, because the duration of the pilot group was only one year and a PET scan is currently not routinely acquired in our hospital prior to an image-guided biopsy, the number of patients in the CT group was approximately twice that in the PET group. In addition, the relative number of bone biopsies obtained from the pelvis was significantly higher in the PET group compared to the CT group. Similarly, the relative number of bone biopsies obtained from the spine was significantly lower in the PET group compared to the CT group. Furthermore, there was a significant difference in the disease profiles between the two groups. This potentially may have influenced the diagnostic success observed with the two guidance procedures. However, our data showed a similar first-try success of ^18^F-FDG-PET-guided biopsy compared to other studies [19,26,27], indicating that our conclusions may not be affected by the imbalances observed. Nevertheless, the results of the present study should be confirmed by a larger prospective study that balances the number of patients between both groups and balances more clinical characteristics, including the biopsy site and disease profiles. Finally, in view of the rising global healthcare costs, we also have to take the additional costs of ^18^F-FDG-PET-guided biopsies into account. Compared to the use of an additional metabolic imaging procedure, such as a PET/CT, to repeating biopsies using CT guidance or surgery for an open biopsy, healthcare costs will increase dramatically (2–5 times higher). However, the additional waiting time in case of an inconclusive biopsy and uncertainty for the patient when repeating a biopsy should be factored in. With the introduction of the new-generation, state-of-the-art PET systems with a higher sensitivity, thereby requiring lower radiopharmaceutical dosages, the trade-off of costs for repeated biopsies will be even lower.

## 5. Conclusions

The increased success rate of first-try biopsies and reduction in inconclusive biopsies demonstrated that ^18^F-FDG-PET/CT imaging prior to a CT-guided biopsy can help to improve the diagnosis of bone lesions. ^18^F-FDG-PET/CT imaging enables the interventional radiologists to isolate lesions that are poorly visible on other imaging modalities and helps them in obtaining conclusive biopsy samples that can be used for histopathological analysis. Prospective research should be performed to confirm our findings.

## Figures and Tables

**Figure 1 cancers-16-02693-f001:**
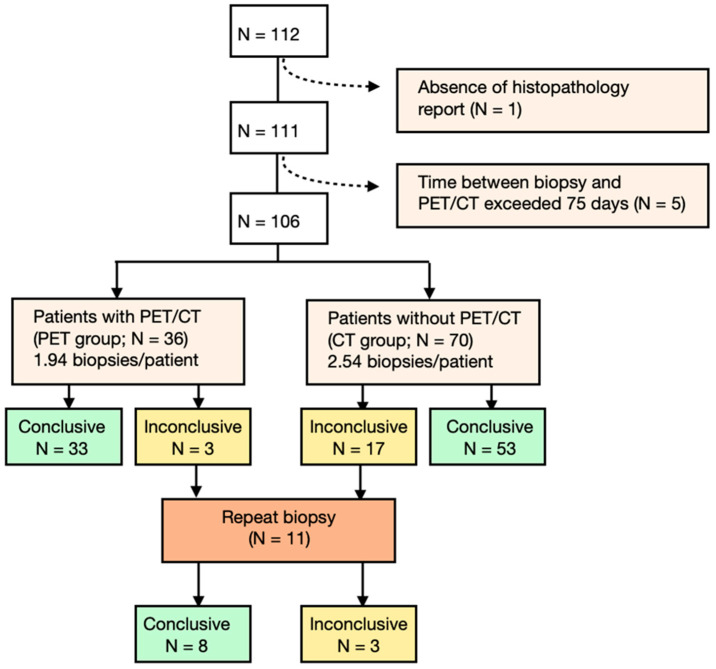
Patient flowchart of the study design.

**Figure 2 cancers-16-02693-f002:**
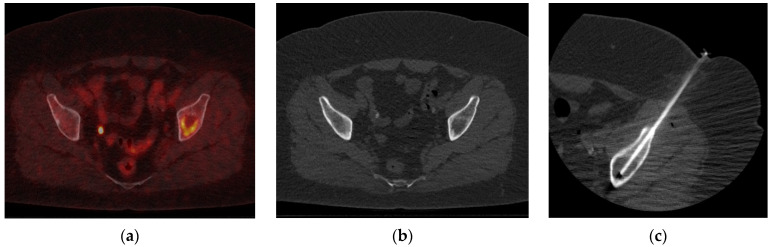
On the left, a trans-axial fusion ^18^F-FDG-PET/CT image of the pelvic area is seen (**a**). There was increased uptake in the dorsal part of the lytic lesion in the os ilium at the left side. In the middle, a corresponding CT with a lytic lesion is presented (**b**). The metabolic activity on the ^18^F-FDG-PET images was peripherally located, with hardly any activity in the center of the lytic lesion. This resulted in a biopsy of the posterior cranial part of the os ilium (**c**), and not the core of the lesion. Histopathology confirmed a metastasis of the primary non-small-cell lung cancer (NSCLC).

**Figure 3 cancers-16-02693-f003:**
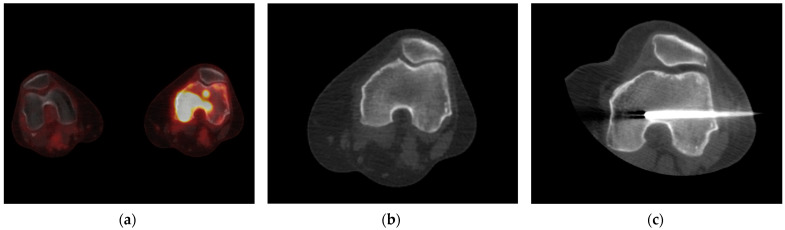
Illustration of the added value of ^18^F-FDG-PET/CT to guide a percutaneous core biopsy when CT does not show any morphologic substrate of a metabolically active lesion. The ^18^F-FDG-PET/CT image shows a bone lesion in the distal femur with evident metabolic activity (**a**). However, the CT (**b**) shows no morphological changes in this patient. The biopsy was performed after the PET/CT (**c**). The histopathology report of the biopsy indicated a malignancy, i.e., a B-cell lymphoma.

**Figure 4 cancers-16-02693-f004:**
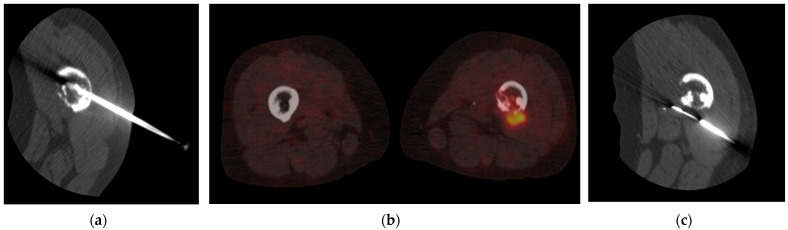
Fluoroscopic images (**a**,**c**) of the area of interest show a lytic lesion and cortical destruction of the bone. No soft tissue lesions are seen. The first-try biopsy (**a**) resulted in an inconclusive histopathologic finding. Fused coronal ^18^F-FDG-PET/CT image of the femur (**b**) shows increased uptake in the left femur but, namely, extension to soft tissue at the medial side of the femur. Panel (**c**) shows the angle and tip of the needle in the second try after performing a metabolic ^18^F-FDG-PET/CT scan. The tip is clearly seen in the soft tissue area, where the ^18^F-FDG-PET scan revealed increased metabolic uptake. The new tract of accessing the tumor was successful and the histopathological findings provided a clear result of the biopsy.

**Table 1 cancers-16-02693-t001:** Clinical baseline characteristics of the included patients.

	PET Group (n = 36)	CT Group (n = 70)	*p*-Value *
Gender (M/F)	20/16	42/28	0.660
Age (years), median (IQR)	66.5 (54–75.75)	61.5 (37–72.25)	0.064
Length (cm), median (IQR)	172 (165–177)	175 (165–183.5)	0.206
Weight (kg), median (IQR)	77 (65–92.85)	74 (67–84.43)	0.462
Number of biopsies (per patient)	1.94	2.54	0.051
Biopsy site, n (%)			**0.008**
- Femur	2 (5.6%)	13 (18.6%)	0.069
- Pelvis	20 (55.6%)	16 (22.9%)	**0.001**
- Costa/sternum	5 (13.9%)	6 (8.6%)	0.395
- Tibia	0 (0%)	7 (10.0%)	-
- Fibula	0 (0%)	3 (4.3%)	-
- Scapula	2 (5.6%)	2 (2.9%)	0.490
- Humerus	3 (8.3%)	4 (5.7%)	0.607
- Spine	2 (5.6%)	15 (21.4%)	**0.035**
- Other	2 (5.6%)	4 (5.7%)	0.973
Indication, n (%)			**0.020**
- Primary malignant	10 (27.8%)	21 (30.0%)	0.812
- Metastatic disease	20 (56.6%)	20 (28.6%)	**0.007**
- Benign disease	2 (5.6%)	17 (24.3%)	**0.017**
- Infection	4 (11.1%)	12 (17.1%)	0.411
Histopathologic outcome, n (%)			
- Primary tumor	8 (22.2%)	33 (47.1%)	**0.013**
- Secondary tumor	24 (66.7%)	22 (31.4%)	**<0.001**
- Osteomyelitis	0 (0%)	8 (11.4%)	-
- Other	4 (11.1%)	7 (10.0%)	0.859

* Bold values denote statistical significance (*p* < 0.05).

**Table 2 cancers-16-02693-t002:** Histopathological analysis from the first biopsy. The number of conclusive and inconclusive first-try biopsies with their histopathological findings, categorized into the subgroups, for both PET and CT groups.

	PET Group (n = 36)	CT Group (n = 70)	*p*-Value *
Conclusive, n (%)	33 (91.7%)	53 (75.7)	**0.047**
- Benign (n = 21)	3 (8.3%)	18 (25.7%)	**0.003**
- Malignant (n = 50)	25 (69.4%)	25 (35.7%)	**<0.001**
- Infection (n = 5)	0 (0%)	5 (7.1%)	-
- Normal (n = 10)	5 (13.9%)	5 (7.1%)	0.260
Inconclusive, n (%)	3 (8.3%)	17 (24.3%)	**0.047**

* Bold values denote statistical significance (*p* < 0.05).

## Data Availability

The datasets generated and/or analyzed during the current study are available from the corresponding author upon reasonable request.

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
