# Peer review of "Augmenting CT-Guided Bone Biopsies Using 18F-FDG PET/CT Guidance"

_cancers, 2024, doi:10.3390/cancers16152693_

Round 1

Reviewer 1 Report

Comments and Suggestions for Authors

General comments

The study compared diagnostic bone biopsies obtained under the guidance by either CT or PET/CT. The study showed that the number of conclusive biopsies was relatively higher when guided by PET/CT than with CT, and that the PET/CT guidance led to more cases where one biopsy only was needed. The manuscript is well written and easy to read. I have only a few specific comments outlined below.

Specific comments

Page 1, lines 35-37

The number of patients with a conclusive biopsy in the PET/CT group was actually lower (33) than in the CT group (53), but the relative number of patients with a conclusive biopsy in the PET/CT (92%) was higher than in the CT group (76%).

In the next sentence “the number of first try biopsies” do you mean: “the average number of biopsies counted for each patient”? (page 5, line 167).

Page 2, line 78

How did you arrive at a time of 75 days and not shorter between the PET/CT scan and obtaining the bone biopsy?

Page 2, line 89

Where is this Department of Radiology located?

Page 5, Table 1

The number of patients in the CT groups is approximately twice that in the PET/CT group. Why did you not extend the duration of the study so that more patients could be included in the PET/CT group?

The relative number of bone biopsies obtained from the pelvis was significantly higher in the PET/CT group than in the CT group. Similarly, the relative number of bone biopsies obtained from the spine was significantly lower in the PET/CT group. In addition, there is a significant difference in the disease profiles between the two groups. Please discuss whether these differences influenced the diagnostic success observed with the two guidance procedures in the Discussion.

Author Response

Reviewer 1:

General comments

The study compared diagnostic bone biopsies obtained under the guidance by either CT or PET/CT. The study showed that the number of conclusive biopsies was relatively higher when guided by PET/CT than with CT, and that the PET/CT guidance led to more cases where one biopsy only was needed. The manuscript is well written and easy to read. I have only a few specific comments outlined below.

We thank the reviewer for his/her positive feedback and thorough review of our manuscript, which helped to improve our manuscript. See below the point-by-point reply to his/her comments.

Specific comments

Page 1, lines 35-37

The number of patients with a conclusive biopsy in the PET/CT group was actually lower (33) than in the CT group (53), but the relative number of patients with a conclusive biopsy in the PET/CT (92%) was higher than in the CT group (76%).

In the next sentence “the number of first try biopsies” do you mean: “the average number of biopsies counted for each patient”? (page 5, line 167).

               ANTWOORD

Page 2, line 78

How did you arrive at a time of 75 days and not shorter between the PET/CT scan and obtaining the bone biopsy?

               ANTWOORD

Page 2, line 89

Where is this Department of Radiology located?

               The hospital name was added to this line as suggested by the reviewer.

Page 5, Table 1

The number of patients in the CT groups is approximately twice that in the PET/CT group. Why did you not extend the duration of the study so that more patients could be included in the PET/CT group?

               ANTWOORD

The relative number of bone biopsies obtained from the pelvis was significantly higher in the PET/CT group than in the CT group. Similarly, the relative number of bone biopsies obtained from the spine was significantly lower in the PET/CT group. In addition, there is a significant difference in the disease profiles between the two groups. Please discuss whether these differences influenced the diagnostic success observed with the two guidance procedures in the Discussion

               ANTWOORD

Reviewer 2 Report

Comments and Suggestions for Authors

Congratulations to Droste et al. on this relevant manuscript. The authors investigated the utilization of bone biopsies guided by 18F-FDG PET/CT. The authors presented a topic that is both pertinent and interesting for readers. The images provided are highly illustrative. However, I have a few remarks to enhance the clarity and rigor of the study.

This manuscript appears to be a cross-sectional study. Please clearly define the study design. Stating that it is a retrospective study is not enough.  

Selection bias is a significant concern in diagnostic studies. Provide a detailed description of the patient selection methods, including the characteristics of the included patients. In addition, detail any other testing that patients underwent.

Explain how the PET/CT scans (index tests) were interpreted. Specify how many individuals were involved in assessing the images. Also, detail the expertise and experience of these professionals.

Describe the interpretation of histological analysis (reference test). Indicate how many people evaluated the histological slides. Also, detail the expertise and experience of these professionals.

The methods section details the time interval between the biopsy and the PET/CT scans. The authors presented this information in the Results section, but it should be cited in the Methods section.

Were the PET/CT results interpreted independently of the histological test results?

Although the figures are informative, the image resolution needs improvement.

Use vs. in italics.

In the Tables footnotes, clarify that the bold p-values denote p < 0.05.

Author Response

Reviewer 2:

Congratulations to Droste et al. on this relevant manuscript. The authors investigated the utilization of bone biopsies guided by 18F-FDG PET/CT. The authors presented a topic that is both pertinent and interesting for readers. The images provided are highly illustrative. However, I have a few remarks to enhance the clarity and rigor of the study.

We thank the reviewer for his/her positive feedback and thorough review of our manuscript, which helped to improve our manuscript. See below the point-by-point reply to his/her comments.

This manuscript appears to be a cross-sectional study. Please clearly define the study design. Stating that it is a retrospective study is not enough.  

Selection bias is a significant concern in diagnostic studies. Provide a detailed description of the patient selection methods, including the characteristics of the included patients. In addition, detail any other testing that patients underwent.

We thank the reviewer for the suggestion, We have stated this is a cross-sectional study in the aim of the Introduction section and in the Patient Demographics subsection (2.1) in the Materials and Methods section. In addition, we added the inclusion criteria to the Patient Demographics subsection (2.1) in the Materials and Methods section.

Explain how the PET/CT scans (index tests) were interpreted. Specify how many individuals were involved in assessing the images. Also, detail the expertise and experience of these professionals.

               ANTWOORD

Describe the interpretation of histological analysis (reference test). Indicate how many people evaluated the histological slides. Also, detail the expertise and experience of these professionals.

               ANTWOORD

The methods section details the time interval between the biopsy and the PET/CT scans. The authors presented this information in the Results section, but it should be cited in the Methods section.

The reviewer is right, this should also be mentioned in the Methods section and is now more explicitly mentioned in the inclusion criteria in the Patient Demographics subsection (2.1) of the Materials and Methods section.

Were the PET/CT results interpreted independently of the histological test results?

               ANTWOORD

Although the figures are informative, the image resolution needs improvement.

               We thank the reviewer for the suggestion and have improved the resolution of all figures.

Use vs. in italics.

               We changed all ‘vs.’ to italics.

In the Tables footnotes, clarify that the bold p-values denote p < 0.05.

We thank the reviewer for the suggestion and added that the bold p-values denote p < 0.05 to Tables 1 and 2.
